

# The destructive subterranean termite *Reticulitermes flavipes* (Blattodea: Rhinotermitidae) can colonize arid territories

David Hernández-Teixidor[1], Aura Pérez-Morín[1], Juan Pestano[2], David Mora[3] and Silvia Fajardo[4]

[1] Island Ecology and Evolution Research Group, Instituto de Productos Naturales y Agrobiología (IPNA-CSIC), La Laguna, Tenerife, Canary Islands, Spain
[2] Tragsatec, La Laguna, Tenerife, Canary Islands, Spain
[3] Anticimex España, Barcelona, Spain
[4] Servicio de Biodiversidad, Consejería de Transición Ecológica, Lucha Contra el Cambio Climático y Planificación Territorial, Gobierno de Canarias, Santa Cruz de Tenerife, Tenerife, Canary Islands, Spain

## ABSTRACT

*Reticulitermes flavipes*, one of the most destructive subterranean termite species, has been detected for the first time in an arid territory: Lanzarote (Canary Islands, Spain). This invasive species was introduced into several countries but never such a dry region. Although there are places with presence of this termite at similar or even higher temperatures, none has annual rainfall (10.1 mm) as low as Lanzarote. On this island it is present in semi-desert, near an affected urban area. Distribution, genetic, climate and host-plant data are evaluated to track and understand its invasion process in the archipelago.

## INTRODUCTION

*Reticulitermes flavipes* (Kollar, 1837) is one of the most damaging subterranean termite pests in both its native and introduced ranges (*Evans, Forschler & Grace, 2013*). Its populations threaten buildings, trees and shrubs around the world, causing severe economic losses (*Khan & Ahmad, 2018*).

This species shares three characteristics with other termites that make them invasive: they feed on wood, have reproductive or potentially reproductive individuals in the wood, and easily generate secondary reproductive individuals (*Evans, Forschler & Grace, 2013*). These characteristics favor its human-mediated dispersal, colony foundation success and rapid spread (*Suppo et al., 2018*; *Eyer & Vargo, 2021*). It has been introduced into Chile, Uruguay, Argentina, and Canada on the American continent and in the following European countries: Austria, Germany, France, Italy, Portugal (Terceira, Azores), and Spain (Tenerife, Canary Islands) (*Aber & Fontes, 1993*; *Austin et al., 2012*, *2005*; *Becker, 1970*; *Carrijo, Battilana & Morales, 2023*; *Ghesini et al., 2010*; *Heisterberg, 1958*;

Corresponding author
David Hernández-Teixidor,
davidhdez@ipna.csic.es

*Ripa & Castro, 2000*; *Scaduto et al., 2012*; *Hernández-Teixidor et al., 2019*). Recently, the invasion history of this species around the world has been documented, concluding that extensive long-distance jump dispersal events occurred frequently in both the native and introduced ranges, presumably through human transportation (*Eyer et al., 2021*).

This invasive termite is able to live within a very wide temperature range, specifically between 3.4 °C and 34.6 °C, showing greater activity at higher temperatures within this range (*Cao & Su, 2016*). In general, termites require high humidity levels (*Rust & Su, 2012*) and prefer moistened soil to build their galleries (*Su & Puche, 2003*). This species can survive in temperate climatic regions where the natural forest habitat is normally cool. This is because anthropogenic factors have promoted an artificial habitat in the cities, characterized by warmer temperature, higher levels of humidity and excess food (*Suppo et al., 2018*). In colder places like Canada, it survives by burrowing deep into the soil to avoid exposure to extreme temperatures (*Clarke, Thompson & Sinclair, 2013*).

*Reticulitermes flavipes* has been present in the Canary Islands (Spain) since 2009, specifically on Tenerife at least (*Hernández-Teixidor et al., 2019*). However, we suspect this species arrived there in the early 2000s. The origin of this introduction remains unknown because the sequences of the Tenerife individuals constitute a unique haplotype and are related to a clade composed of sequences from multiple sources within its native and introduced distributional range (*Hernández-Teixidor et al., 2019*). It was detected for the first time in northeast Tenerife, which is currently the main affected area. Subsequently, several other populations were found relatively near the main area and also in two distant touristic/commercial localities in the southwest of the island. In addition, it was detected recently on another island of this archipelago, Lanzarote. This island has much drier conditions than Tenerife. Can this introduced termite adapt to life in such arid areas?

In this article, we present the first report of *R. flavipes* for the island of Lanzarote, providing distribution, genetic, climatic and host plant data. Its possibilities of expansion in an arid region are discussed.

## MATERIALS AND METHODS

### Sampling area

This study took place in the Canary Islands, a volcanic archipelago comprising seven major islands off the south-west Atlantic coast of NW Africa. Although we are attentive to the appearance of subterranean termites on any island, our active search has been on Tenerife since 2017 and since 2020 also on Lanzarote (Fig. 1).

The climate of the Canary Islands is classifiable as maritime subtropical according to their latitudinal location and oceanic character, and as mediterranean owing to a seasonal regime with summer drought. However, there are important climatic variations due to altitude/elevation and orientation, influenced by the trade-winds and the cool Canary Current (*del Arco & Rodríguez, 2018*). Tenerife is the highest and largest island of the Canaries, situated near the centre of the archipelago. Due to its altitude reaching through and above the NE trade-wind clouds, the climate is more humid on its windward north

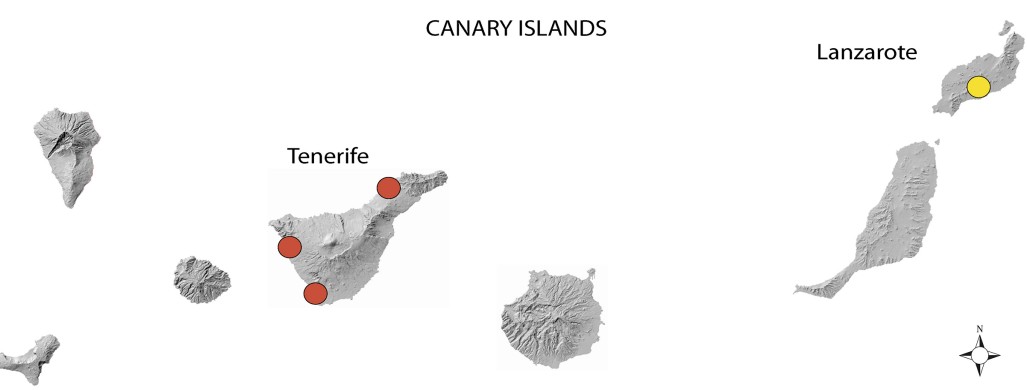

CANARY ISLANDS

Lanzarote

Tenerife

**Figure 1 Localities with a presence of *Reticulitermes flavipes* in the Canary Islands.** Those on Tenerife are represented as red disks and on Lanzarote as yellow ones. Map source credit: Government of the Canary Islands, CC-BY 4.0.               

and northeast slopes, and drier to the south and west (*Villa et al., 2003*). Lanzarote is one of the driest areas in the European Union, and the most arid island in the Canaries (*Tejedor, Jiménez & Díaz, 2003*). Due to its comparatively low elevation, the trade-winds generate much less mist and condensation on its slopes. It is usually therefore without the benefit of cloud cover (*del Arco & Rodríguez, 2018*).

The distribution data for this termite species on the islands derive initially from occasional samples and data provided by pest control companies and citizens, mainly by active and passive searching since 2017. Active sampling consisted of a visual inspection looking for sure signs of the presence of this species, such as shelter tubes and wood damage both inside buildings and outside on trees and other timber. In areas where active sampling did not detect termites, we undertook passive sampling with wooden stakes for further detection. More than 2,500 points with the presence of *R. flavipes* were used to delimit the current distribution area in the Canaries, 70 of them on Lanzarote (Figs. S1 and S2). Distribution areas and distances between localities were measured and represented using QGIS 3.4.10 (Open Source Geospatial Foundation).

In addition, the area occupied by this species in the Canary Islands from the first record (2009) to 2022 and its expansion over time were calculated for each year using QGIS. A concave enveloping contour procedure was applied to delimit the area of the different populations. Subsequently, a buffer of 30 m was applied to extend the influence of the termites beyond the points that form the edge of the polygon. To follow the progress of the species from year to year, linear distances were calculated in all directions between the area it occupied each year. To do this, points were marked along the boundaries of the polygons defining the areas of occurrence and the shortest lines drawn between the points of the areas in consecutive years.

On Lanzarote, specimens of all castes were collected throughout the small area where this invasive termite was detected and studied. Samples were stored in ethanol 100%, and two of them sequenced to confirm species identification. The Gobierno de Canarias granted permission for collecting material (N. 2021/13873).

### DNA data

DNA from two workers from Lanzarote was extracted using Qiagen DNeasy Blood and Tissue Kit (Qiagen, Hilden, Germany), amplifying cytochrome oxidase II using TL-J-3037 (TED)/TK-N-3785 (EVA) primers (*Simon et al., 1994*). Sequences obtained were MAFFT aligned with sequences from Tenerife (*Hernández-Teixidor et al., 2019*) and with reference sequences of the genus *Reticulitermes* from GenBank, using the program Geneious 8.0.5.

### Climatic data

The climate data were taken from the meteorological stations nearest the areas showing presence of these subterranean termites in both islands (Fig. S3) (*AEMET, 2021*; *Agrocabildo de Tenerife, 2021*; *Agrocabildo de Lanzarote, 2021*; *Gobierno de Canarias, 2021*; *MAPA, 2021*). Data from three stations were used for Lanzarote (GC102, LZ01 and LZ airport) and six for Tenerife (TF105, TF106, TF108 and TF109 from North Tenerife; TF02 and TF13 from South Tenerife) (Table S1). Climatic variables selected were "mean temperature", "maximum temperature", "minimum temperature", "mean humidity", and "precipitation" between the years 2015–2019.

These data were statistically analyzed applying generalized linear models (GLM) with the *lme4* package of R 4.1.1 (*R Core Team, 2021*), to explore differences between the *R. flavipes* populations from each island (Islands: Lanzarote-Tenerife) and populations grouped by zones (zones: North Tenerife (NT), South Tenerife (ST) and Lanzarote (LZ)). When comparing zones, the variables were first analyzed regardless of seasons, and later checked for possible differences linked to seasonal conditions. We used one model for each variable, considered as dependent variables, and Islands and zones as factors, assuming a Gaussian distribution to fit the GLM. In addition, paired t-tests were performed between zones to establish which showed more similar climatic conditions.

To determine if Lanzarote is one of the most arid areas where *R. flavipes* has been detected, we visually compared the climatic data for the different regions where it has been reported. We used the annual mean temperature (19.31 °C–20.50 °C), annual max. temperature (23.90 °C–24.92 °C), annual min. temperature (14.75 °C–16.08 °C) and annual precipitation (9.83–12.50 mm) from Worldclim version 2.1, with a spatial resolution of 30 s ($\approx 1 \text{ km}^2$) (*Fick & Hijmans, 2017*). We selected the values of the climatic variables in the area of Lanzarote where termites are found and highlighted the areas with similar values around the world. Given that precipitation is the main factor limiting the expansion of this termite, we considered the regions where all variables coincided or where precipitation overlapped with at least one temperature variable. In addition, we indicate the geographic areas that are warmer and drier than Lanzarote and also the regions with an annual mean temperature range that this species can tolerate (3 °C–30 °C) with low precipitation.

## RESULTS

*Reticulitermes flavipes* is reported for Lanzarote, the second island recorded for the Canary archipelago. This new affected area is about 270 km away from the main colonized focus on Tenerife. So far, this termite species was detected only in a central-eastern area of

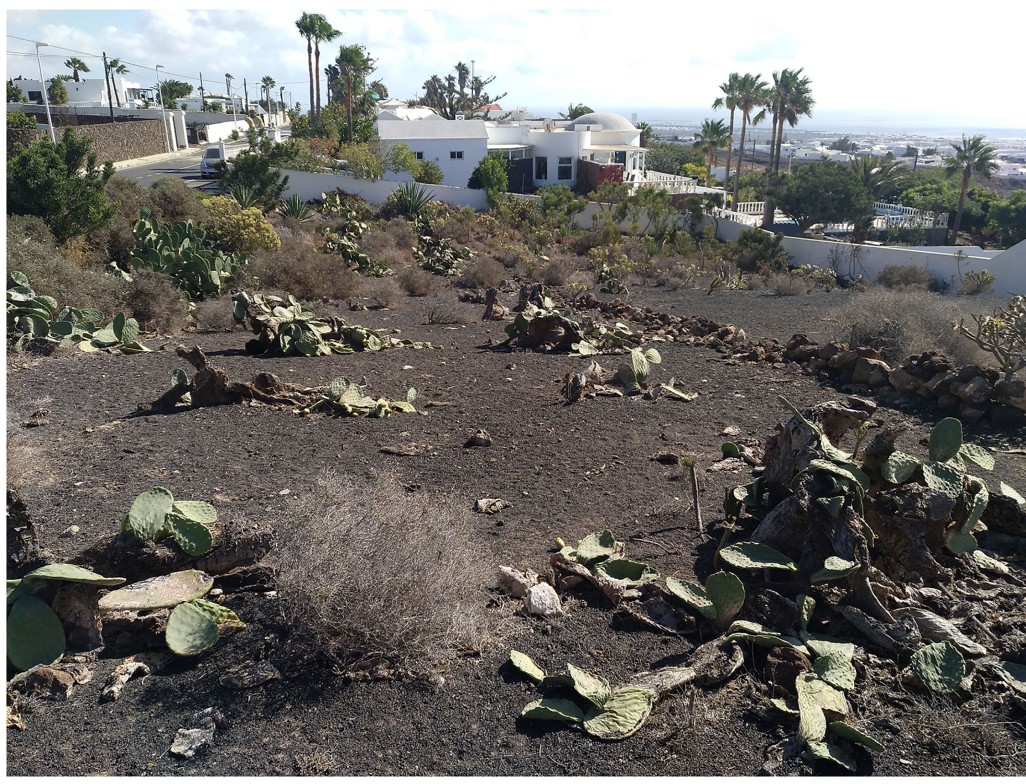

**Figure 2 Overview of the area of Lanzarote affected by *R. flavipes*.** Photo credit: Silvia Fajardo.

Lanzarote called Güime (San Bartolomé; 28.97132, −13.6089) at the end of 2019. To date, it is affecting buildings, gardens, small crops, unbuilt urban plots and abandoned agricultural land, totalling a surface area of at least 21,000 m² on the island (Fig. 2).

The 649-bp fragments obtained confirm the identity of the species, they are 100% identical to *R. flavipes* sequences from Tenerife (MH794537–MH794541).

On Tenerife, the present distribution comprises an estimated area of 3.829 km². Currently, there is one large primary focus in the northeast of the island (between the coordinates 28.53226, −16.40063 to the north, 28.49035, −16.41848 to the south, 28.52721, −16.38411 to the east and 28.50871, −16.42062 to the west). Several populations are 3–4 km away, with two more distant sites in the south, approximately 60 km away (Las Américas: 28.05756, −16.731, Puerto Santiago: 28.235, −16.837; Fig. S3). This subterranean species is affecting buildings, crops and both native and ornamental plant species, with consequent economic, social and biodiversity impacts.

Since its first detection in 2009, this termite has expanded to occupy the current distribution (The annual growth of each population can be seen in Tables S2 and S3). In 2020, the area grew enormously in Tenerife: 1,538,870 m² of the main area, plus the new populations detected away from it. In the following two years, growth was considerable but declining (Table S2). On Lanzarote, growth was more gradual, although in the last year, it almost doubled (Table S3). Linear distances of *R. flavipes* advance in the primary focus ranged from 40 to 434 m on average, with maximum values exceeding 1,000 m between

**Table 1 Differences in climatic variables between Lanzarote and Tenerife.**

**Islands**

| | $\chi^2$ | df | $p$ | Mean ± SE Lanzarote | Mean ± SE Tenerife |
|---|---|---|---|---|---|
| **Mean temperature (°C)** | 33.256 | 1 | $8.08 \times 10^{-9}$ | 20.5 ± 0.210 | 19.0 ± 0.147 |
| **Max. temperature (°C)** | 10.434 | 1 | 0.001237 | 28.5 ± 0.325 | 27.2 ± 0.228 |
| **Min. temperature (°C)** | 37.189 | 1 | $0.072 \times 10^{-9}$ | 15.3 ± 0.233 | 13.6 ± 0.163 |
| **Mean humidity (%)** | 4.7675 | 1 | 0.029 | 70.1 ± 0.571 | 68.6 ± 0.399 |
| **Precipitation (mm)** | 25.785 | 1 | $3.817 \times 10^{-7}$ | 10.1 ± 2.17 | 23.6 ± 1.52 |

**Table 2 Differences in climatic variables between the three zones.**

**Zones (NT-ST-LZ)**

| | $\chi^2$ | df | $p$ | Mean ± SE NT | Mean ± SE ST | Mean ± SE LZ |
|---|---|---|---|---|---|---|
| **Mean temperature (°C)** | 99.189 | 2 | $2.2 \times 10^{-16}$ | 18.2 ± 0.171 | 20.5 ± 0.241 | 20.5 ± 0.199 |
| **Max. temperature (°C)** | 37.635 | 2 | $6.72 \times 10^{-9}$ | 26.4 ± 0.272 | 28.8 ± 0.385 | 28.5 ± 0.317 |
| **Min. temperature (°C)** | 66.274 | 2 | $4.06 \times 10^{-15}$ | 13.0 ± 0.195 | 14.7 ± 0.276 | 15.3 ± 0.227 |
| **Mean humidity (%)** | 33.096 | 2 | $6.50 \times 10^{-8}$ | 70.0 ± 0.477 | 65.6 ± 0.674 | 70.1 ± 0.557 |
| **Precipitation (mm)** | 101.92 | 2 | $2.2 \times 10^{-16}$ | 32.1 ± 1.75 | 6.37 ± 2.47 | 10.13 ± 2.03 |

**Note:**
NT, North Tenerife; ST, South Tenerife; LZ, Lanzarote.

2010–2022 (Fig. S4). Small populations close to the main area showed smaller values (10–150 m), similar to those present in south Tenerife (12–213 m) but more widely spread than those on Lanzarote (10–30 m) (Fig. S5; Tables S4 and S5).

In the Canary islands, this invasive termite has been detected in living ornamental trees and shrubs (*Acacia* sp., *Araucaria* sp., *Bismarckia nobilis*, *Cestrum nocturnum*, *Coccoloba uvifera*, *Cupressus* sp., *Delonix regia*, *Dypsis lutescens*, *Echinocactus grusonii*, *Euphorbia pulcherrima*, *Ficus elastica*, *Ficus microcarpa*, *Gossypium* sp., *Hibiscus* sp., *Howea forsteriana*, *Monstera deliciosa*, *Pelargonium* sp., *Phoenix canariensis*, *Rosa* sp., *Schefflera actinophylla*, *Schinus molle*, *Washingtonia filifera*, *Yucca* sp.), fruit trees (*Carica papaya*, *Citrus × sinensis*, *Eriobotrya japonica*, *Ficus carica*, *Mangifera indica*, *Passiflora edulis*, *Persea americana*, *Prunus domestica*, *Prunus persica* and *Vitis vinifera*), introduced naturalized shrubs (*Foeniculum vulgare*, *Nicotiana glauca*, *Opuntia* spp., *Ricinus communis*), other crops (*Beta vulgaris* var. *cicla*, *Ipomoea batatas*, *Phaseolus vulgaris*, *Solanum tuberosum*) and native plant species (*Artemisia thuscula*, *Euphorbia lamarckii*, *Dracaena draco*, *Kleinia neriifolia*, *Rumex lunaria*). In addition, *R. flavipes* has been seen feeding on fallen fruit of the following domestic species: *Citrus reticulata*, *Malus domestica*, *Mangifera indica*, *Persea americana*, *Prunus persica* and *Pyrus communis*.

The climatic data results show significant differences in all variables between Lanzarote and Tenerife (Table 1). The mean values of all climatic variables except precipitation are higher for Lanzarote (Table 1). When we specifically compare the three zones of the Canaries where termites are present, significant differences appeared in all the climatic variables (Table 2).

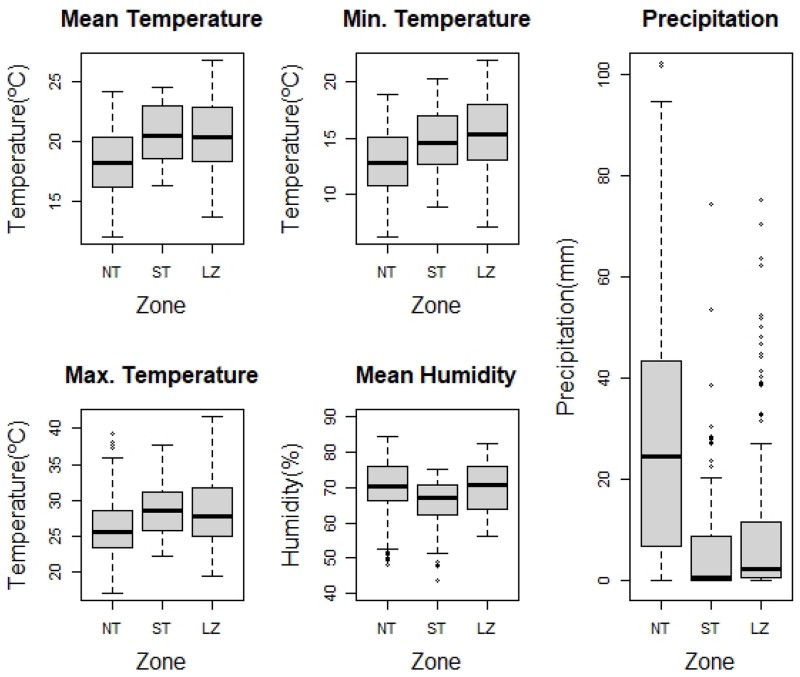

**Figure 3 Differences in climatic variables between the three affected zones.** The boxes comprise 50% of the data, the black horizontal lines representing the median. The whiskers extend to the last value within 1.5 times the interquartile range. Abbreviations: NT, North Tenerife; ST, South Tenerife and LZ, Lanzarote.  

Considering the pairwise comparisons (Table S6), South Tenerife and Lanzarote do not show significant differences in climatic variables (Fig. 3), except in mean humidity ($z = -4.43673$; $p = 0.0001$). Additionally, when climatic variables were analyzed seasonwise, significant differences were detected between all of them (Table S7), except mean humidity in autumn (NT: 70.1%, ST: 67.4%, LZ: 70.8%. Pairwise comparisons (Table S8) showed that South Tenerife and Lanzarote were very similar in all the seasons. However, there were some significant differences in min. temperature in summer (NT: 15.4 °C, ST: 17.3 °C, LZ: 18.4 °C), and max. temperature (NT: 23.6 °C, ST: 26.3 °C, LZ: 24.0 °C) and mean humidity (NT: 40.32%, ST: 58.1%, LZ: 38.3 °C) in winter (Fig. S6).

The geographic areas worldwide with similar annual climatic values to the area affected by this species in Lanzarote are located in North Africa, the Middle East, southwest Africa, western North and South America, and central Australia (Fig. 4). Specifically, the most similar regions to Lanzarote are southern Morocco, northern Egypt, Libya and Algeria, north-eastern Somalia, south-eastern Iran and Yemen, north-west Namibia, western Peru, western United States (*e.g.*, Las Vegas) and Mexico (Fig. 4). When we consider the regions where this invasive termite is present, only Lanzarote shows these climatic values. Warmer and drier regions than Lanzarote include the Sahara Desert, Middle East, southwest Namibia, northern Baja California and western Peru (Fig. S7), but when we consider the range of temperatures at which this species survives (3–30 °C) at the same precipitation levels, regions such as northeast China and northern Chile are also comparable (Fig. S8).

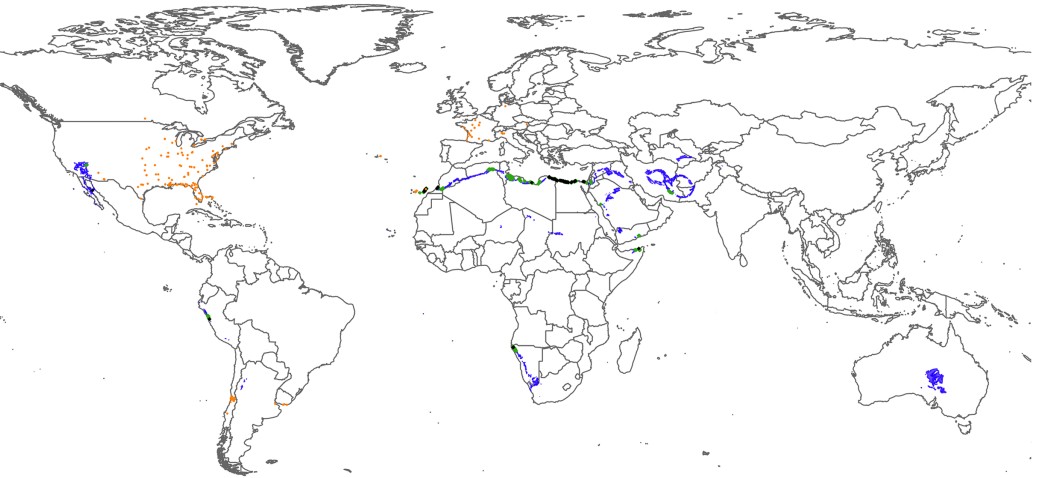

**Figure 4 World map representing the zones with four (black), three (green) or two (blue) of the studied climatic variables collated for Lanzarote and the distribution of *Reticulitermes flavipes* (orange dots).** Map data © EuroGeographics for the administrative boundaries.

## DISCUSSION

The genetic results confirmed that the subterranean termite present on Lanzarote is *Reticulitermes flavipes* and its populations derive from the same introduction as Tenerife (*Hernández-Teixidor et al., 2019*). This termite species has been on Lanzarote since more than nine years ago, when the first inadequate treatments were applied to affected properties due to an incorrect identification of the species (*Gobierno de Canarias, 2020*). The owners of the first house affected on the island commented that their garden was landscaped with plants from nurseries located in parts of Tenerife affected by this termite. Together with the genetic data, the greater age and surface area of the Tenerife colonies support this hypothetical introduction route. They are believed to have been present on Tenerife since the beginning of the 2000s.

The area occupied by *R. flavipes* in the Canary Islands has grown steadily since its first detection to a considerable area. This increase has been due to the species' own (natural) and human-mediated dispersal, probably for more than 20 years. However, this data is inflated due to increased sampling efforts after 2020, which has led to detecting areas where this cryptic species was not known to occur. It can be naturally spread by alates that can fly a few hundred meters and propagate by neotenics. The latter can actively leave their parent colonies accompanied by workers (*i.e.*, budding) or be part of a colony fragment passively carried away by human transport (*Suppo et al., 2018*). There is little information on the dispersal rates of this or other invasive termite species worldwide. In France, *Suppo et al. (2018)* used a *R. flavipes* dispersal rate of about ten m/year in colonies older than five years. In that country, *R. flavipes* is mainly spread by neotenics, since alates are relatively rare (*Dronnet et al., 2005*; *Baudouin et al., 2018*). In the Canaries, alates may play a more decisive role than in colder areas such as France, although it seems that neotenics are mainly responsible for dispersal. The annual progress distances of this invasive termite in

the Canary Islands were longer than expected, probably due to the detection of new but previously existing undetected areas. In smaller populations such as Lanzarote, the annual advance distances were shorter and can be explained by natural dispersion.

Temperature is a major factor influencing the distribution of many species (*Doucet, Walker & Qin, 2009*), including *R. flavipes* whose abundance is also affected (*Vargo, Juba & Deheer, 2006*). Although it can live at a wide range of temperatures (*Cao & Su, 2016*), all known populations of this termite, native and introduced, are located in temperate zones. In many regions where termites were introduced, its presence is closely associated with urban environments. For instance, the colony established in Germany was able to survive due to the urban heating systems (*Sellenschlo, 1988*). Where there are climatic barriers, the termite's capacity to burrow deep within the soil can facilitate colony establishment (*Clarke, Thompson & Sinclair, 2013*) by buffering temperature variations, even without the additional protection offered by anthropization of their habitat. In cold areas, its nests can be 75–95 cm below the surface to avoid low temperatures (*Husby, 1980*). In warm dry areas like Lanzarote, this species may also burrow deep into the soil to withstand high temperatures.

The characteristics that make termites invasive (*Evans, Forschler & Grace, 2013*) provide them with a competitive advantage. Climate change can increase the suitability of habitat, improving species survival and facilitating reproduction, thus promoting population persistence (*Walther et al., 2009*). In this context, a 16.7% increase in the global distribution of *R. flavipes* is expected by 2050, enabling it to colonize colder areas further north (*Buczkowski & Bertelsmeier, 2017*). In regions such as the Canary Islands, when the temperature rises, its food consumption rate will be higher and therefore its expansion and resultant damage can advance faster.

The Canary Islands have a subtropical climate with relatively constant mild temperatures year-round. This stability facilitates the establishment of many introduced species and their expansion throughout the archipelago, as occurs with other social insects (*Hernández-Teixidor et al., 2020*). However, Lanzarote is the most arid island of the archipelago (*Tejedor, Jiménez & Díaz, 2003*) and its habitats should not be so appropriate for this subterranean species. As expected, examination of the variables shows Lanzarote has more arid conditions than Tenerife, with higher mean, maximum and minimum temperatures, higher mean humidity and less precipitation. Its moderate elevation does not permit moisture discharge from the trade winds, therefore the overall climate is dry and similar to the southern slopes of the more mountainous islands of the archipelago, such as Tenerife (*Tejedor, Hernández-Moreno & Jiménez, 2007*). Lanzarote and south Tenerife are very similar in all the climatic variables except mean humidity, which is lower in the two south Tenerife populations, probably due to their closeness to the coast. Comparing these climatic variables with other regions inhabited by this species, some such as the Bahamas, south Florida and Texas are warmer than Lanzarote. However, when rainfall is considered, Lanzarote is the only place with such low annual precipitation where this termite is present. Given that this variable is very important for *R. flavipes* to form its soil galleries (*Su & Puche, 2003*) and above-surface mud tubes, the climatic conditions of Lanzarote can be considered harsh for the species because it is one of the driest areas where

*R. flavipes* is found. Considering that this termite is able to survive and expand in these conditions, there are some *a priori* unfavorable regions that it could be colonized, such as north Africa or Peru. Moreover, warmer and drier regions could also be inhabited by this species if transported by human beings to an anthropized locality that mitigates the prevailing climatic conditions.

In many regions where this invasive species is present, it is mainly associated with urban areas. In north Tenerife, *R. flavipes* is present in urban and rural areas but also in semi-naturalized and natural areas, feeding on wild plant species. The climatic conditions of this area, with warm mean temperatures (18.2 °C) and high mean humidity (70.1%) and precipitation (32.1 mm), allow this species to settle and spread throughout the territory, without it being urbanized. However, in south Tenerife, this termite has only been found to date in buildings and gardens with irrigation, which provides it with favorable artificial conditions to survive in such a dry territory. On Lanzarote it has also mainly been located in houses and irrigated gardens, but is furthermore able to colonize adjacent semi-desert areas, under even more unfavorable climatic conditions.

On Tenerife, *R. flavipes* has been spreading through the soil, occupying a considerable area in the northeast of the island, besides being dispersed over medium or long distances by human-mediated transport (*Hernández-Teixidor et al., 2019*). The arid conditions of Lanzarote could be a limiting or retardant factor for its natural expansion. However, due to transport of trees, wood, *etc.*, it could already have unknown populations in other parts of Lanzarote.

In the Canary Islands this species is now well established and undergoing a clear process of demographic and spatial expansion, considering the surface area occupied, distance between localities, varied habitats, climatic conditions, and the number of plant species it consumes on the two islands.

These new distribution data suggest that this invasive termite may have a wider distribution than known to date in the archipelago, facilitated by people transporting infested material over long distances within and between islands. Probably, its introduction onto Lanzarote was through infested plants from nurseries in Tenerife, more than nine years ago (*Gobierno de Canarias, 2020*). For this reason, preventing the introduction of potentially invasive species is the most effective strategy to avoid damage and the costs involved in control actions. If prevention and biosecurity measures fail and an alien species is introduced, early detection and a rapid response protocol should be carried out (*Waugh, 2009*; *Reaser et al., 2020*). To halt the spread of *R. flavipes* on Tenerife, an eradication program has been implemented since 2020. The main objectives are to: 1) establish preventive and control measures to avoid its spread beyond its distribution area, 2) design a protocol for early detection and rapid response to new focal colonies of the species, 3) determine its current range on the island, 4) control and eradicate its populations, and 5) implement a monitoring programme to ensure eradication is as effective as possible. To date, 262,000–354,500 m$^2$ have been baited with 0.5% hexaflumuron at 8060 stations (*Cabildo de Tenerife, 2022*). Currently, a management and eradication project has just started on Lanzarote with the same aims as Tenerife.

## CONCLUSIONS

*Reticulitermes flavipes*, a subterranean termite from temperate zones, has been detected in one of the driest areas in the European Union: the island of Lanzarote in the Canary Islands. This surprising finding opens the possibility that this invasive species can establish itself and thrive in arid regions that initially did not meet its requirements. It is therefore also necessary to intensify the measures to avoid inadvertently transporting this species to dry regions and to mitigate possible damage it causes after establishment.

The intrinsic characteristics of the species that have allowed it to become dominant and highly invasive in many temperate regions around the world, along with the ongoing anthropization of many areas, seem to facilitate its survival in drier regions.

## ACKNOWLEDGEMENTS

We are grateful to Daniel Suárez and Eduardo Jiménez-García (IPNA-CSIC), who sequenced the specimens from Lanzarote, and to Gerardo Garcia (Tragsatec) for his help with climatic data analysis. Staff from Tragsatec, IPNA-CSIC, the Canary Government, RedExos, and Anticimex participated in the fieldwork. The Canary Government Biodiversity Service, and the Cabildos (island councils) of Tenerife and Lanzarote provided support and collecting permits. The manuscript was edited by Guido Jones.

### Funding

This study was supported by funds from Cabildo de Tenerife and Gobierno de Canarias *via* Tragsatec. David Hernández-Teixidor was funded by the Cabildo de Tenerife, under the TFinnova Programme supported by MEDI and FDCAN funds. The funders had no role in study design, data collection and analysis, decision to publish, or preparation of the manuscript.

### Grant Disclosures

The following grant information was disclosed by the authors:
Cabildo de Tenerife and Gobierno de Canarias *via* Tragsatec.
MEDI and FDCAN funds.

### Competing Interests

Juan Pestano is employed by Tragsatec and David Mora is employed by Anticimex España.

### Author Contributions

- David Hernández-Teixidor conceived and designed the experiments, performed the experiments, prepared figures and/or tables, authored or reviewed drafts of the article, and approved the final draft.
- Aura Pérez-Morín analyzed the data, prepared figures and/or tables, authored or reviewed drafts of the article, and approved the final draft.

- Juan Pestano performed the experiments, authored or reviewed drafts of the article, and approved the final draft.
- David Mora performed the experiments, authored or reviewed drafts of the article, and approved the final draft.
- Silvia Fajardo performed the experiments, authored or reviewed drafts of the article, and approved the final draft.

### Field Study Permissions

The following information was supplied relating to field study approvals (*i.e.*, approving body and any reference numbers):

Permission for collecting material was granted by the Gobierno de Canarias (N. 2021/13873).

### DNA Deposition

The following information was supplied regarding the deposition of DNA sequences:

The sequence is available in the Supplemental File and at GenBank: OQ988003.

### Data Availability

The raw data is available in the Supplemental File.

### Supplemental Information

Supplemental information for this article can be found online at http://dx.doi.org/10.7717/peerj.16936#supplemental-information.

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
