# Peer review of "The destructive subterranean termite Reticulitermes flavipes (Blattodea: Rhinotermitidae) can colonize arid territories"

_PeerJ, doi:10.7717/peerj.16936_

## Round 0.1 · original submission · Major Revisions

Please revise your paper, paying special attention to the concerns of the reviewers. In particular, you need to justify the use of only two samples.

Reviewer 1 ·

Basic reporting

The clarity of the English writing in this article is commendable, and there is adequate background information provided. However, the overall information presented is limited, as only two samples were collected and sequenced. The analysis of climatic data relied on general information about the islands rather than specific data from each collection site. As a result, more detailed results cannot be presented, such as the route of termite introduction and the establishment of microenvironments.

Experimental design

This article serves as a scientific report or case study, highlighting a significant finding in the field of termite management or invasive biology, rather than general biology. The primary data presented in this study consists of two collections of Reticulitermes flavipes in Lanzarote.

Validity of the findings

Given that the same team previously reported the presence of R. flavipes in Tenerife, Canary Islands, Spain, the recent discovery of this termite species on another island may not come as a surprise. R. flavipes is known to be a pest in construction and has established itself in manmade structures and garden landscapes. Therefore, analyzing the overall climatic data across the Canary Islands may not accurately reflect the specific microenvironment in which this invasive pest has established itself.

Additional comments

In general, the data supporting the discovery of R. flavipes in Lanzarote is strong. Nevertheless, in order to reach a robust conclusion regarding whether R. flavipes is adapted to very dry climatic zones, it is essential to gather additional samples from various collection sites across these islands and acquire detailed climatic data specific to these microenvironments.

·

Basic reporting

Clear and unambiguous, professional English used throughout.
Literature references, sufficient field background/context provided
Professional article structure, figures, tables. Raw data shared

Experimental design

Original primary research within Aims and Scope of the journal.
Research question well defined, relevant & meaningful. It is stated how research fills an identified knowledge gap.
Methods are not described with sufficient detail and information to replicate. For example line 78 says occasional samples were collected, how was this done? Explain current active searching ie how specimens were obtained. Did you examine all trees in the area? All buildings? Or you depended on reported cases? How did you choose the stations? Termites were collected from 3 stations in Lanzarote, 6 at Tenerife North and 2 from Tenerife south. Did the choice of stations depend on where termites were found or where climatic data was obtained? How did you choose your stations? In line 83, where the individuals collected soldiers, workers or nymph? What % concentration of ethanol was used to preserve specimens? Did you collect specimens from galleries, nests or mud tubes? Buildings and plants were examined for signs of termite presence - mud-tubes, nests or hollow sound or mounds.
In the results section, line 118 gives measurements of distance. This is not in methods. Where and how was measurements taken? Indicate that distance was measured.
Methods for results presented in line 125 to 129 not described
Line 121 shows 8.5 km2 of surface area of termite infestation, how did you measure this. It’s not in methods.


Fig 1 map should have coordinates so that each georeferenced station is indicated in relation to other parts of the country

Validity of the findings

Some results presented appear like discussion. Eg lines 167 to 177 is not part of the results of this study. To compare study with others in a tabular form , references of authors should be added.

Statistically sound study however I suggest you estimate the rate of spread from 2009 to 2023 in islands. For instance in the map of area indicate point of initial presence of termite, note new areas of infestation and estimate rate of spread on the island.
The Figs showing globally favorable climatic conditions for termite invasion are quite interesting. Please indicate if R. flavipes is found in all areas with these favorable conditions.
Conclusions appear well stated and linked to original research question

Additional comments

There should be consistency in reference style Eg if three authors in line 27 are Evans, Forschler & Grace, 2013 then Tejedor et al., 2007 ( line 216) should read Tejedor, Hernandez-Moreno & Jimenez 2007.

Lines 143 to 166. Reads like a repetition of the tables S2 to S6. All the chi square values and df May not be in the text

Reviewer 3 ·

Basic reporting

1) The English needs to be cleaned up a bit to improve the flow of the paper.
2) I was a little surprised not to see a mention of how the termite originally came to be introduced within the Canary Islands. I would imagine from somewhere within the European continent, but maybe it would be worthwhile just to show something like a haplotype network that incorporates samples from here and across the world.
3) It can be pretty difficult to parse a piece of text containing many parentheses and statistical information (at least for me). Since the main text does not already contain a lot of figures (and no tables), I think it could be nice to cut out some of the statistical information present within the text of the results section and instead incorporate some of the supplementary tables as main text tables.
4) To go with the point above, I might also include one of the global maps present within the supplementary information as a main text figure, just to give the reader a more direct visual representation of the similar climatic regions across the world (not everyone will want to open the supp material or go to google maps).

Experimental design

Nothing stands out as flawed to me in terms of the models tested, but I would not consider myself particularly well-versed in this area so I will leave that to the purview of the other reviewers.

Validity of the findings

The findings appear well-supported to me, but again, I am not an expert in regards to modeling. I will say (as I stated above) that I would like to see the link between the invasiveness of this termite (which is a big emphasis of the paper) and the potential origin of the Canary Islands invasion more fleshed out, which I think would connect it more with the literature surrounding this termite and create a more comprehensive paper.

---

## Round 0.2 · accepted · Accept

Thank you for your careful revision of your manuscript.

·

Basic reporting

.

Experimental design

.

Validity of the findings

.

Additional comments

I have re-reviewed the manuscript on "The destructive subterranean termite Reticulitermes flavipes (Blattodea: Rhinotermitidae) can colonize arid territories".
The authors have done a good job. All corrections are satisfactory.
I would only suggest that lines 125 to 135 in the Materials and Methods be written in the third person. Eg instead of ‘we examined the trees’, the authors should say ‘the trees were examined’. This will follow the style of preceding paragraphs.